# Hierarchical Contrastive Learning for Enzyme Function Prediction

**Soorin Yim** [1]  **Doyeong Hwang** [1]  **Kiyoung Kim** [1]  **Sehui Han** [1]

## Abstract

Enzymes are biological catalysts with numerous industrial applications, and they are categorized by the Enzyme Commission (EC) number system based on their catalytic activities. With over 200 million protein sequences identified, experimental characterization of enzymes is impractical, necessitating computational methods. Current approaches face challenges with class imbalance and intrinsic hierarchy of the EC number system. This study employs hierarchical contrastive learning for EC number prediction, effectively integrating the EC number hierarchy into the model. Our approach addresses severe class imbalance and improves prediction performance, particularly for higher hierarchical levels and previously unseen EC numbers, demonstrating enhanced robustness and outperforming existing methods.

## 1. Introduction

Enzymes are specialized proteins that act as biological catalysts, accelerating biochemical reactions crucial for life. They are widely used in various sectors, including food, pharmaceutical, and energy industries (Chapman et al., 2018), driving active research to discover or engineer enhanced enzymes. The Enzyme Commission (EC) number system is a hierarchical classification system that categorizes enzymes based on the reactions they catalyze, with four levels of increasing specificity that describe their function (Tipton & McDonald, 2018). Figure 1 provides an overview and an example of the EC number system.

As of 2023, over 200 million protein sequences have been identified (Consortium, 2022). However, experimentally characterizing each one to determine its function is impractical. Consequently, only a small fraction of these proteins has been experimentally identified as enzymes with specific EC numbers, highlighting a pressing need for computational research in this field.

[1]LG AI Research, Seoul, Republic of Korea. Correspondence to: Sehui Han <hansse.han@lgresearch.ai>.

*Accepted at the 1st Machine Learning for Life and Material Sciences Workshop at ICML 2024.* Copyright 2024 by the author(s).

Deep learning has revolutionized the field of EC number prediction, enabling the accurate prediction of EC numbers for proteins based on their sequences. Initial approaches focused on training classification models, either using separate models for each hierarchical level or by employing multi-task learning, where each hierarchical level was treated as a distinct task (Sureyya Rifaioglu et al., 2019; Ryu et al., 2019; Sanderson et al., 2023). However, these models face challenges with severe class imbalance. As of April 2022, approximately 25.9% of 5,242 EC numbers are annotated by a single protein, each of which is reviewed and reported to Swiss-Prot (Appendix Table 1) (Consortium, 2022; Yu et al., 2023). To address this imbalance, recent research has adopted supervised contrastive learning (CL) (Memon et al., 2020; Yu et al., 2023). Supervised CL involves training a model that represents proteins with the same EC number closely, while ensuring that proteins assigned to different EC numbers are distinctly separated. This technique effectively handles class imbalance, particularly for EC numbers associated with a limited number of known enzymes (Yu et al., 2023). However, previous CL-based approaches did not account for the inherent hierarchy of EC numbers.

EC numbers are continuously added as new enzyme reactions are discovered (IUBMB). While computational methods cannot predict novel EC serial numbers, they can help infer an enzymes function through higher-level class predictions. Therefore, accurately predicting higher-level EC numbers not present in the training set is crucial.

In this paper, we employ hierarchical CL for EC number prediction, explicitly incorporating EC number hierarchy into our model. Our contributions are as follows:

- We demonstrate that hierarchical CL improves EC number prediction performance, particularly for higher levels of hierarchy and for EC numbers that were not present in the training set.

- Our results indicate that hierarchical CL enhances the robustness of EC number prediction.

- Our results show that hierarchical CL outperforms state-of-the-art (SOTA) EC number prediction methods.

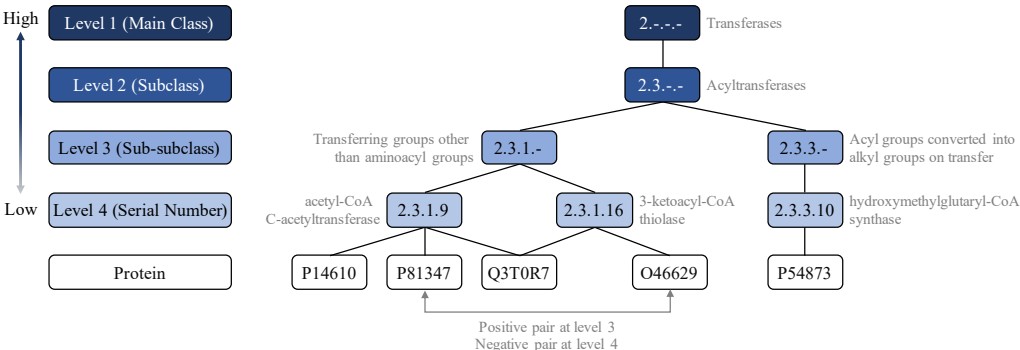

*Figure 1.* Overview of the Enzyme Commission (EC) number system, a hierarchical system for classifying enzymes based on the chemical reactions they catalyze. Depiction of a segment within the EC number hierarchy, where lower levels denote more specific reactions.

## 2. Methods

The latest research on EC number prediction applied vanilla supervised contrastive loss (Khosla et al., 2020) on EC number at lowest level. However, such approaches do not fully leverage hierarchical nature of EC numbers. We propose to use labels across all levels with hierarchical contrastive learning frameworks.

### 2.1. Hierarchical Contrastive Learning

We employed Hierarchical Multi-label Contrastive (HMC) loss (Zhang et al., 2022) as the loss function of our contrastive learning setting. HMC loss extends supervised contrastive loss by applying it to all hierarchical levels.

Denote a hierarchical level as $l \in L$. Then, the pair loss between an anchor sample indexed by $i$ and the positive sample indexed by $p$ at level $l$ is defined as follows:

$$\mathcal{L}^{\text{pair}} = \log \frac{exp(f_i \cdot f_p^l/\tau)}{\sum_{a \in Ai} exp(f_i \cdot f_a/\tau)} \quad (1)$$

where $A(i)$ is set of indexes in a batch except for $i$, and $\tau$ is temperature parameter. Then HMC loss is defined as follows:

$$\mathcal{L}^{\text{HMC}} = \sum_{l \in L} \frac{1}{|L|} \sum_{i \in I} \frac{-\lambda_l}{|P_l(i)|} \sum_{p_l \in P_l} \mathcal{L}^{pair}(i, p_l^i) \quad (2)$$

where the penalty weight $\lambda_l$ differs by each level in the hierarchy, $P_l$ is the set of positive samples at level $l$ for anchor sample indexed by $i$. Basically, by leveraging all levels in hierarchy, HMC loss encourages pairs that share the ancestors more get closer. This enables model to output embedding vectors that can preserve label hierarchy within the embedding space. Additionally, positive pairs that share hierarchy at certain level can be negative pairs at lower level and become "hard negative" samples, which reduces the

need for hard negative mining. Penalty weight $\lambda_l$ could be tuned to optimal point that accomplish best performance. $\lambda_l$ were initially proposed to be functions proportional to the $l$, but we used a list of fixed scalar values since detailed tuning was needed.

In (Zhang et al., 2022), both HMC and Hierarchical Constraint Enforcing (HiConE) loss are used. HiConE loss enforces hierarchy by constraining the loss between pairs from a higher level never become larger than the loss between pairs from a lower level. However, in our experiments, the constraints from HiConE loss was too strong that the model lose its distinguishing power at lower levels.

## 3. Experiments

### 3.1. Experimental Setup

#### 3.1.1. DATASETS

We utilized EC number annotations from Swiss-Prot, processed by (Yu et al., 2023). Specifically, we used split30 and split50 datasets, where protein sequences were split based on sequence identity such that no enzymes in the test set share more than 30%, 50% sequence identity with any enzymes in the training set, respectively. The dataset was split into 5 folds, stratified by their labels, to perform 5-fold cross validation. We used 80%, 10%, 10% as training, validation, and test sets, respectively.

We evaluated the models using two independent test sets from (Yu et al., 2023). The first test set, NEW-392, comprises 392 proteins released after the creation of the training dataset (April 2022). The second test set, Price-149, contains 149 proteins that are challenging due to incorrect or inconsistent labeling by automated annotation methods (Price et al., 2018). Detailed statistics for the datasets are provided in Appendix Table 1.

### 3.1.2. MODEL ARCHITECTURE

To embed protein sequences, we employed ESM-2-650M (Lin et al., 2023), a pretrained protein language model trained on 65 million unique sequences, which leverages transformer architectures to capture evolutionary relationships and predict protein structures. To represent proteins from the ESM embeddings, we used a multi-layer perceptron (MLP) with three hidden layers and layer normalization, following the approach in (Yu et al., 2023). For detailed architectural specifications and hyperparameters, please refer to the Appendix B and Appendix Table 2.

### 3.1.3. INFERENCE

Inference was conducted based on the distance between EC number and the query protein. First, we obtained the embedding of each EC number as the center of the embeddings of proteins associated with that EC number in the training set. Then, we used the Euclidean distance between the EC number embedding and the query protein embedding for inference. For higher-level EC numbers, we assumed that if an enzyme belongs to a specific child EC number (e.g. X.Y.Z.-), it should also belong to the corresponding parent EC number (e.g. X.Y.-.-). Therefore, we used the shortest distance to any child EC number as the distance to the parent EC number. To binarize labels based on the distance, we applied maximum separation method used in (Yu et al., 2023), which selects EC numbers that are most separated from others. For evaluation, we only considered EC numbers present in the training set, as we cannot obtain centers for EC numbers that are absent in the training set.

### 3.1.4. BENCHMARK MODELS AND EVALUATION METRICS

We evaluated our model against three benchmark models: (1) ESM-2: Vanilla ESM-2-650M without contrastive learning. (2) ESM-2+MLP: Conventional multi-task learning setup where we fine-tuned ESM-2-650M with a 2-layered MLP head for predicting level 4 EC numbers using (Zhu et al., 2022). (3) CLEAN: The recent state-of-the-art model utilizing supervised CL without considering EC number hierarchy (Yu et al., 2023). CLEAN employs hard negative mining to improve performance. As the original CLEAN model was trained on ESM-1b, we retrained the model using ESM-2-650M. We evaluated the models in terms of Area Under Precision-Recall Curve (AUPRC), and F1 score at each hierarchical level (levels 1 to 4). Given the multiple classes at each level, we calculated weighted average of AUPRC and F1 scores, with weights based on the number of true instances for each class.

## 3.2. Results

Here, we report the performance of models trained on split30 dataset; performance on the split50 dataset can be found in Appendix D.2. Unless specified otherwise, the results are from cross-validation.

### 3.2.1. HMC FOR LEVEL 3 AND LEVEL 4

We initially applied HMC to level 3 and level 4, with the performances illustrated in Figure 2. When HMC is applied solely to level 4 (denoted as 'Level3:Level4 ratio of 0:1'), it functions identically to vanilla supervised CL. This approach implicitly captures the EC number hierarchy, as indicated by its performance on higher-level EC numbers. Conversely, applying HMC exclusively to level 3 (denoted as '1:0') results in superior performance for higher-level EC numbers but performs poorly for predicting level 4 EC numbers. This is because it treats different level 4 EC numbers under the same level 3 EC numbers as identical.

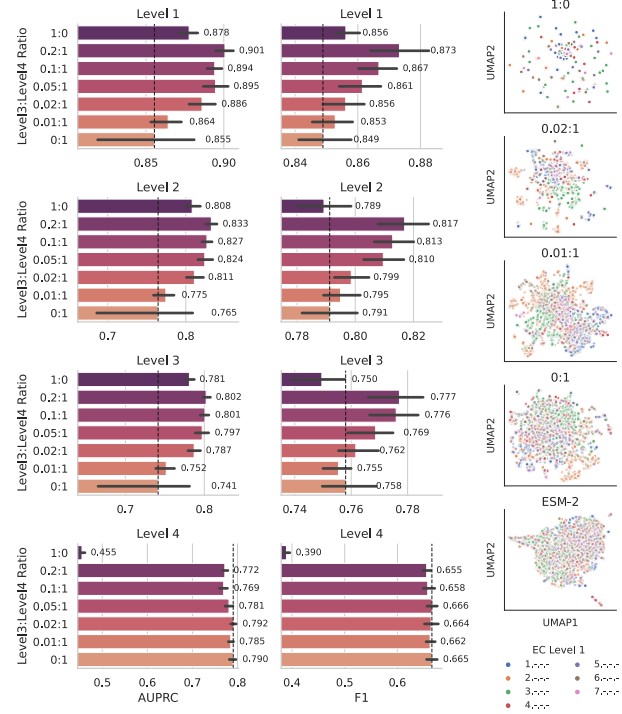

*Figure 2.* Performance of HMC for EC number level 3 and level 4. The figure shows AUPRC (left) and F1 score (middle) for HMC with varying weights $\lambda_l$ for level 3 and level 4, with stronger $\lambda_3$ indicated by darker colors. Each row represents the performance for predicting each level of the EC number hierarchy. The vertical line indicates the performance of HMC applied solely to level 4. (right) UMAP visualization of level 4 EC number embeddings, with EC numbers are colored based on their level 1 class.

To address this, we conducted a grid search to determine

the optimal weight for level 3 ($\lambda_3$) while fixing the weight for level 4 ($\lambda_4$) at 1. As shown in Figure 2, increasing $\lambda_3$ reinforces the hierarchy but can compromise performance at level 4. Nonetheless, we could identify an optimal $\lambda_3$ that improves performance for both level 3 and level 4.

We visualized the embeddings of level 4 EC numbers in Figure 2. The results indicate that vanilla ESM-2 embeddings do not effectively capture the EC number hierarchy. However, vanilla supervised CL ('0:1') implicitly learns this hierarchy, with EC numbers clustered according to their level 1 class. As we increase the weight to level 3 ($\lambda_3$), EC numbers with common parents form tighter clusters. HMC applied only to level 3 ('1:0') squashes different EC numbers into the same embeddings, failing to distinguish them effectively.

### 3.2.2. EXTENSION OF HMC TO ALL LEVELS

We extended HMC to encompass all levels of EC number hierarchy, from level 1 to level 4. The weight for each level ($\lambda_l$) is selected based on a incremental grid search from lower levels to higher levels. We selected weights with the highest AUPRC at level 4, then proceeded to the next level, as described in Appendix D.1.

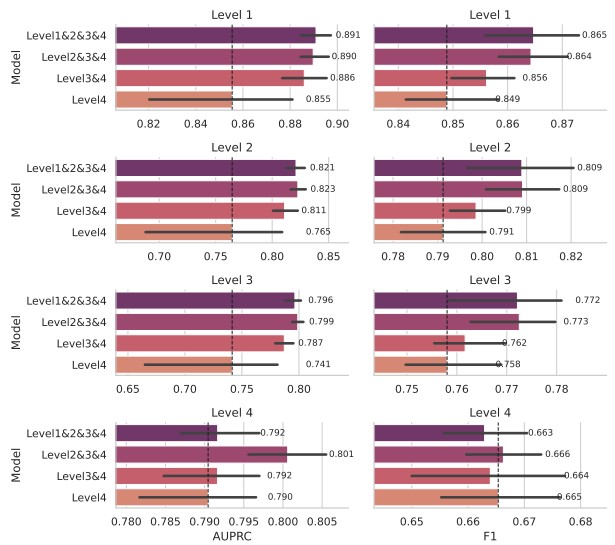

*Figure 3.* Performance of HMC from level 1 to level 4. Performance for predicting each level of the EC number hierarchy is shown in each row. The vertical line indicates the performance of HMC applied exclusively to level 4.

Our results indicate that by gradually incorporating the hierarchical structure into the HMC loss, we can enhance performance at higher levels while maintaining performance at lower levels, as shown in Figure 3.

### 3.2.3. PREDICTION OF UNSEEN EC NUMBERS

Predicting level 4 EC numbers for entirely new enzymes with previously undefined EC numbers is infeasible. However, higher-level predictions, such as level 3, are possible. Accurately predicting the higher-level EC numbers is crucial, in order to be confident on model's prediction over new proteins. Figure 4 illustrates the performance at level $l$ for proteins with level $(l + 1)$ labels unseen in training dataset. Our results show that incorporating higher levels in the HMC loss significantly improves the performance. This is particularly effective for unseen or sparse labels, which accounts for more than quarter of the level 4 EC numbers (Appendix Table 1).

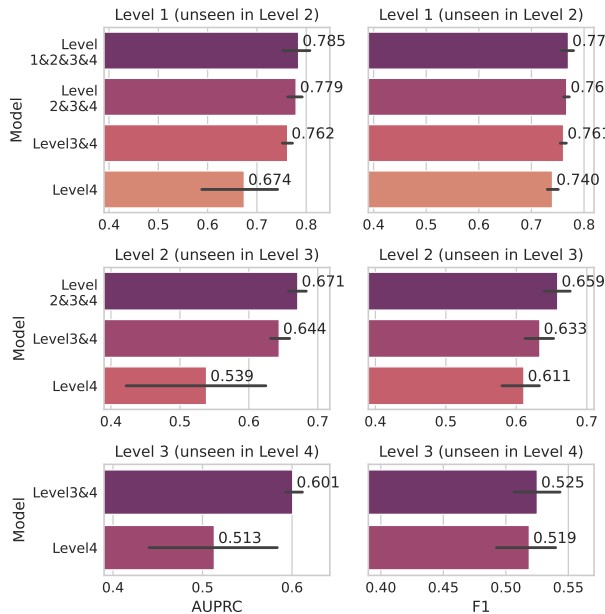

*Figure 4.* The performance of HMC on unseen EC numbers. Each row shows performance on level $l$ with instances whose label at level $(l + 1)$ is unseen in training set.

### 3.2.4. EVALUATION AGAINST BENCHMARK MODELS

In Figure 6, we compared the performance of our models to three benchmark models mentioned in 3.1.4. Overall, our model 'Level1&2&3&4' and 'Level2&3&4' achieved SOTA F1 score on New-392 and Price-149 datasets, across different levels. Additionally, model 'Level4' shows better F1 score than CLEAN overall. We suspect that the performance gap is due to usage of hard negative mining, which was analyzed with detail in (Robinson et al., 2020).

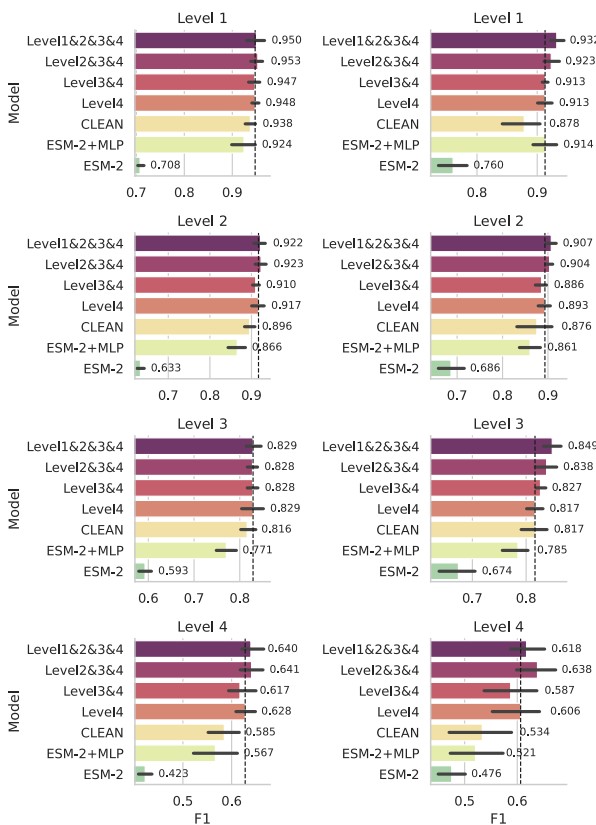

Figure 5. Performance comparison to benchmark models. (left) Performance on New-392 dataset (right) Performance on Price-149 dataset. Vertical line indicates the performance of HMC applied solely on level 4.

### 3.2.5. ROBUSTNESS

In Figure 6, we counted the number of proteins with matching predictions across all 5-fold models in New-392 and Price-149 dataset. This measure indicates model robustness. Our model achieves the highest consensus, with gradual increase as more hierarchical level is incorporated. If representations are clustered according to the hierarchical structure, less errors will occur on descendent prediction, benefiting from ancestor information. Thus above result demonstrate that hierarchical contrastive learning improves robustness.

### 3.2.6. CASE STUDY FOR IDENTIFYING THE FUNCTION OF A NEW ENZYME

This case study demonstrates the effectiveness of incorporating EC number hierarchy in predicting the function of a new enzyme. A0A1D8PH52, from the NEW-392 dataset, belongs to 2.3.1.9 (acetyl-CoA C-acetyltransferase). A0A1D8PH52 was initially misclassified as 2.3.3.10

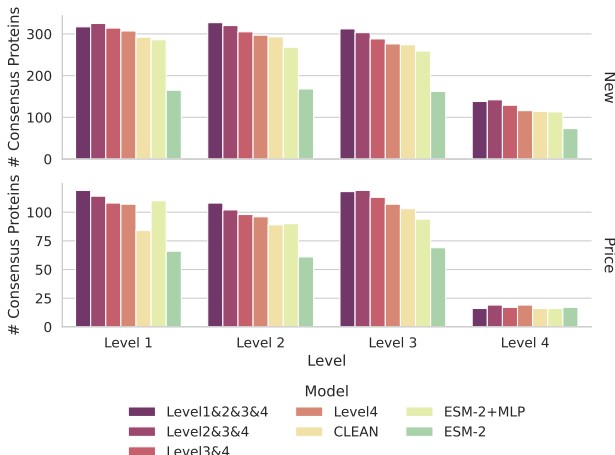

Figure 6. Robustness of prediction results. The number of consensus proteins for which all 5 models from cross validation predicted identical EC numbers.

(hydroxymethylglutaryl-CoA synthase) when HMC loss was applied solely on level 4. Although the model correctly predicted the level 2 EC number, 2.3.-.- (Acyltransferases), it failed at level 3. By considering the hierarchy of EC numbers, the enzyme was accurately identified as 2.3.1.9.

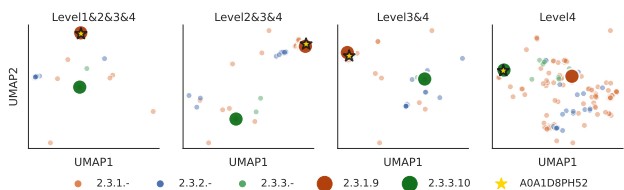

Figure 7. Case study for identifying the function of the enzyme A0A1D8PH52 from the NEW-392 dataset. Initially misclassified as 2.3.3.10 (hydroxymethylglutaryl-CoA synthase), incorporating EC number hierarchy correctly identifies it as 2.3.1.9 (acetyl-CoA C-acetyltransferase).

## 4. Conclusion

This study demonstrates the effectiveness of hierarchical contrastive learning in EC number prediction. Our results show that employing the hierarchical nature of EC number improves the model performance and robustness over SOTA models, particularly for higher level EC numbers and previously unseen EC numbers. These results highlight the potential of hierarchical CL to advance the field of computational enzyme annotation, providing a robust and scalable solution for the accurate prediction of enzyme functions.

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
