## A. Dataset Statistics

As described in Section 3.1.1, we used protein sequences reviewed by experts and reported to Swiss-Prot (Consortium, 2022). This dataset, denoted as 'split100', was curated by CLEAN (Yu et al., 2023) and divided into two subsets: split30 and split50. We used split30 and split50 as training sets, while NEW-392 and Price-149 served as independent test sets. Table 1 details the number of proteins in each dataset and the number of EC numbers at each hierarchical level.

One of the challenges in EC number prediction is the sparse annotation of many classes, especially for level 4. Table 1 shows that in split100 dataset, each level 4 EC number has an average of 46 positive proteins out of 227,362. Additionally, more than a quarter of level 4 EC numbers have only one positive protein, indicating severe class imbalance. Furthermore, novel proteins may belong to previously undefined EC numbers that are unseen to the model.

## B. Model Architecture

All of the models used identical architecture, which consists of encoder network that maps feature to initial representation using pretrained model, and projection network that maps initial representation to final representation vectors which are ready for use in inference phase. We used pretrained ESM-2-650M (Lin et al., 2023) model as encoder network that acquires initial representation $r \in \mathcal{R}^{D_E}$ from the sequence of protein, with weight parameters frozen. Projection network then maps $r$ into final representation $f \in \mathcal{R}^{D_o}$, which is acquired by following equations:

$$f = W_3(\text{Mod}_2(\text{Mod}_1(r)))  \quad (3)$$
$$\text{Mod}_i(\cdot) = \text{ReLU}(\text{Dropout}(\text{LayerNorm}(W_i(\cdot))))  \quad (4)$$

Linear matrix $W_1 \in \mathcal{R}^{D_E \times D_h}, W_2 \in \mathcal{R}^{D_h \times D_h}, W_3 \in \mathcal{R}^{D_h \times D_o}$ are trained to get task-appropriate representation $f$.

## C. Hyperparameters

Hyperparameters are given in table 2. We trained models were trained up to 3000 epochs, and the model with the lowest validation loss was selected for testing.

## D. Experimental Results

### D.1. Weight Grid Search for Levels 2 and 1

We conducted a grid search to optimize weights for each hierarchical level ($\lambda_l$), starting with level 3 and progressing to levels 2 and 1, while keeping the $\lambda_4$ fixed at 1. For each level, we selected the weight with the highest AUPRC from cross-validation. For level 3, $\lambda_3$ of 0.02 yielded the highest AUPRC for level 4 (Figure 2). Consequently, we fixed $\lambda_3$ and $\lambda_4$ at 0.02 and 1, respectively, and proceeded to optimize the $\lambda_2$, as shown in Figure 8.

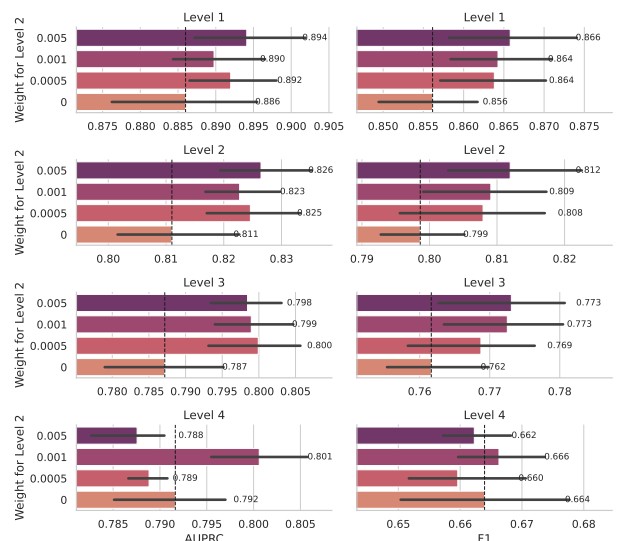

*Figure 8.* Grid search on the weight for level 2, $\lambda_2$

With $\lambda_2$ of 0.001 achieving the highest AUPRC for predicting level 4 EC numbers, we fixed $\lambda_2, \lambda_3, \lambda_4$, at 0.001, 0.02, and 1, respectively. We then conducted a grid search to optimize $\lambda_1$, as depicted in Figure 9.

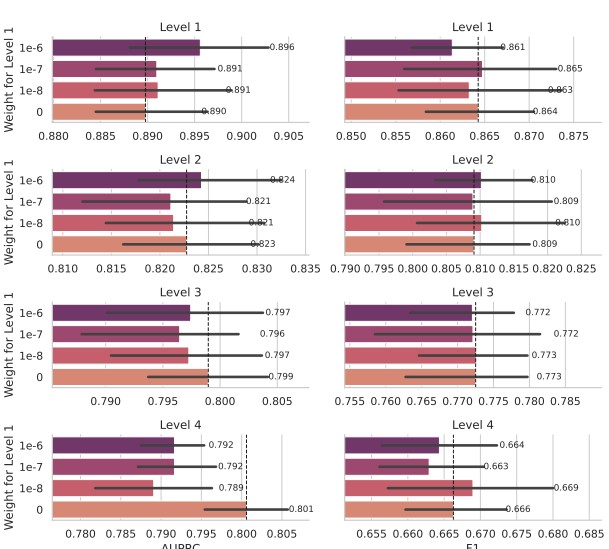

*Figure 9.* Grid search on the weight for level 1, $\lambda_1$

*Table 1.* Dataset statistics.

| DATASET | # PROTEINS | LEVEL | # EC NUMBERS | # PROTEIN-EC NUMBER PAIRS | AVG. PROTEINS PER EC NUMBERS | # SINGLE-PROTEIN EC NUMBERS |
|---------|-----------|-------|-------------|---------------------------|------------------------------|------------------------------|
| SPLIT30 | 10,202 | LEVEL 1 | 7 | 10,427 | 1,490 | 0 |
|         |        | LEVEL 2 | 72 | 10,504 | 146 | 6 |
|         |        | LEVEL 3 | 240 | 10,555 | 44 | 26 |
|         |        | LEVEL 4 | 3,576 | 11,033 | 3 | 2,051 |
| SPLIT50 | 29,942 | LEVEL 1 | 7 | 30,543 | 4,363 | 0 |
|         |        | LEVEL 2 | 72 | 30,772 | 427 | 3 |
|         |        | LEVEL 3 | 251 | 30,916 | 123 | 16 |
|         |        | LEVEL 4 | 4,709 | 32,027 | 7 | 2,168 |
| SPLIT100 | 227,362 | LEVEL 1 | 7 | 232,306 | 33,187 | 0 |
|         |        | LEVEL 2 | 72 | 233,878 | 3,248 | 1 |
|         |        | LEVEL 3 | 253 | 234,855 | 928 | 7 |
|         |        | LEVEL 4 | 5,242 | 241,025 | 46 | 1,360 |
| NEW-392 | 392 | LEVEL 1 | 7 | 396 | 57 | 0 |
|         |     | LEVEL 2 | 28 | 397 | 14 | 4 |
|         |     | LEVEL 3 | 57 | 399 | 7 | 17 |
|         |     | LEVEL 4 | 177 | 503 | 3 | 89 |
| PRICE-149 | 149 | LEVEL 1 | 6 | 149 | 25 | 0 |
|         |     | LEVEL 2 | 19 | 149 | 8 | 4 |
|         |     | LEVEL 3 | 27 | 149 | 6 | 8 |
|         |     | LEVEL 4 | 56 | 152 | 3 | 27 |

*Table 2.* Hyperparameters for model architecture and training.

| | HYPERPARAMETER | VALUE |
|---|---|---|
| MODEL ARCHITECTURE | DROPOUT | 0.3 |
| | $D_E$ | 1280 |
| | $D_h$ | 512 |
| | $D_o$ | 256 |
| TRAINING | BATCH SIZE | SPLIT30: 6000, SPLIT50: 10000 |
| | OPTIMIZER | ADAMW |
| | LEARNING RATE | 5.00E-04 |
| | WEIGHT DECAY | 1.00E-03 |
| | CONTRASTIVE LOSS TEMPERATURE $\tau$ | 0.1 |

The optimized weights for each level ($\lambda_l$) are illustrated in Figure 10. As the number of unique EC numbers decreases logarithmically, the weights for each hierarchical level also decrease logarithmically.

### D.2. Experimental Results on Split50 Dataset

Figure 11 shows performances on grid search for the weight on level 3 ($\lambda_3$), starting with weight for level 4 ($\lambda_4$) as 1. Similar to performance on split30 dataset, there exists an optimal point of weight $\lambda_3$ and $\lambda_4$ for performance on level 3 and level 4. Raising weight for level 3 $\lambda_3$ gives slight performance gain in level 3 while performance on level 4 is not lost. Figure 12 shows performances on split50 dataset when extending HMC to all levels. $\lambda_l$ weights follows that from split30. Extending HMC to higher levels gives

comparable or slightly better performance on all levels, most noticeable when extended to level 3. We conclude that effectiveness of hierarchical CL can be generalized on larger EC number datasets.

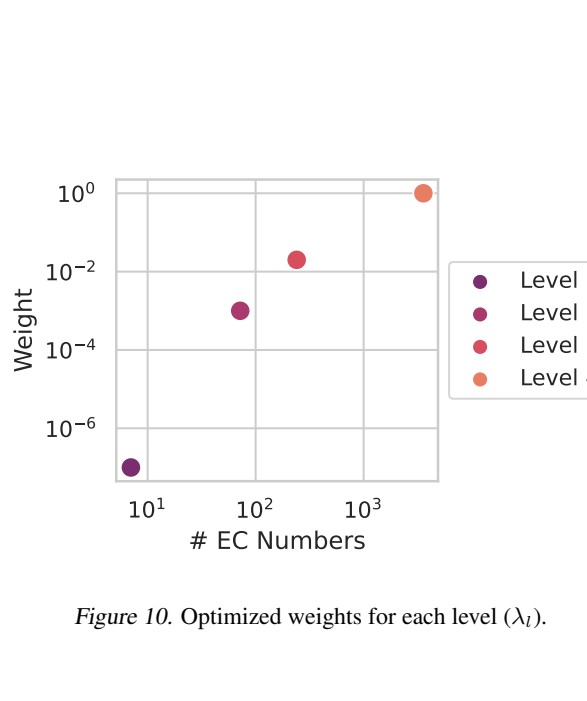

Figure 10. Optimized weights for each level ($\lambda_l$).

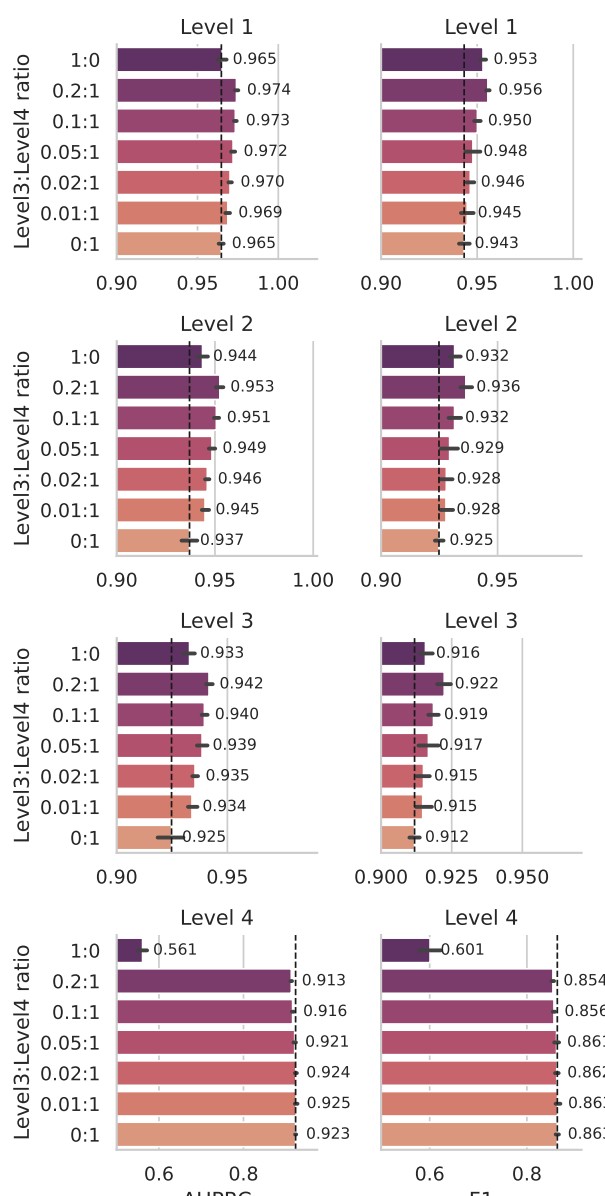

Figure 11. HMC for level 3 and level 4 in split50 dataset.

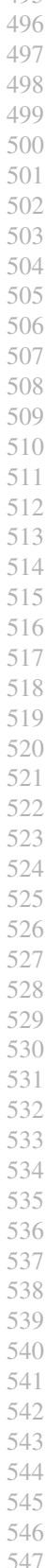
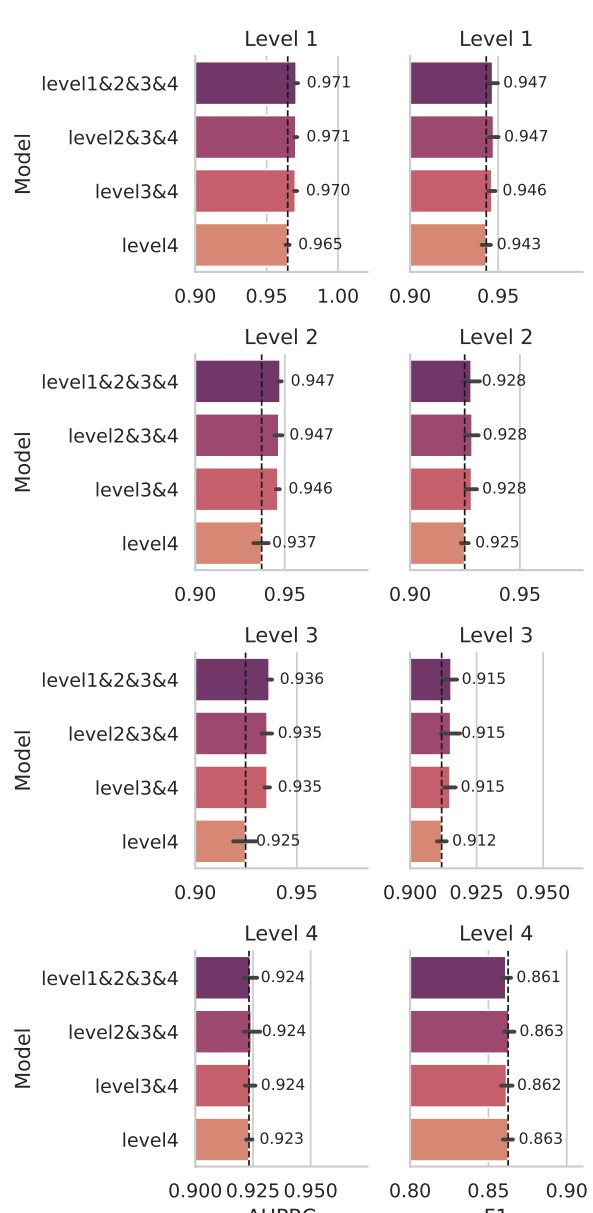

*Figure 12.* HMC from level 1 to level 4 in split50 dataset. Extending HMC for higher levels improves the model performance.