# OpenReview forum: "Hierarchical Contrastive Learning for Enzyme Function Prediction"
_ICML.cc/2024/Workshop/ML4LMS — ML4LMS Poster_

### Official Review · Reviewer_W9PB · 2024-06-11
**Review for Hierarchical Contrastive Learning for Enzyme Function Prediction**

**Rating:** 8
**Confidence:** 4

**Review:**

Quality - 3/5
Clarity - 4.5/5
Originality - 3/5
Significance - 3/5

---

### Official Review · Reviewer_xKXh · 2024-06-12
**The paper advance SOTA with hierarchical contrastive learning approach for enzyme function prediction, effectively addressing class imbalance and hierarchical integration.**

**Rating:** 8
**Confidence:** 3

**Review:**

Pros:
- Incorporation of hierarchy in the contrastive learning approach to predict EC numbers, which produce more accurate and robust predictions particularly for higher hierarchical levels and unseen EC numbers.
- (Hierarchical) contrastive learning handles the extreme class imbalance in the EC system.
- Extensive ablation study.
- The method outperformed SOTA (CLEAN), though CLEAN relied on older protein embedding (ESM1b vs ESM2). The author noted that  Level4 or Level3&4 ratio 0:1 function identically to vanilla CL, which means that CLEAN trained on ESM2 should have comparable result to these levels instead.

Cons:
- Lacking (experimental or other) validation that SOTA method paper had.
- The case study of 1 enzyme is nice but it would have been more useful to do a systematic study of misclassification when using only CL instead of hierarchical CL.